

# Martian magnetism with orbiting sub-millimeter sensor: Simulated retrieval system

Richard Larsson[1], Mathias Milz[2], Patrick Eriksson[3], Jana Mendrok[3], Yasuko Kasai[1], Stefan Alexander Buehler[4], Catherine Diéval[5], David Brain[6], and Paul Hartogh[7]

[1]National Institute of Information and Communications Technology, Tokyo, Japan
[2]Luleå University of Technology, Kiruna, Sweden
[3]Chalmers University of Technology, Göteborg, Sweden
[4]University of Hamburg, Germany
[5]Lancaster University, UK
[6]University of Colorado, Boulder, US
[7]Max Planck Institute for Solar System Research, Göttingen, Germany

*Correspondence to:* Richard Larsson (ric.larsson@gmail.com)

**Abstract.** A Mars-orbiting sub-millimeter sensor can be used to retrieve the magnetic field at low altitudes over large areas of significant planetary crustal magnetism of the surface of Mars from measurements of circularly polarized radiation emitted by the 368 GHz ground-state molecular oxygen absorption line. We design a full retrieval system for one example orbit to show the expected accuracies on the magnetic field components that one realization of such a Mars satellite mission could achieve. We find that the two horizontal components of the magnetic field can be measured at about 200 nT accuracy, globally, with a vertical resolution of about 4 km from about 6 km up to 70 km in tangent altitude. The method and some of its potential pitfalls are described and discussed.

## 1 Introduction

In the past decades, there have been several proposals to fly a sub-millimeter sensor on a satellite mission to Mars. One such proposal is to fly the Far-InfraRed Experiment, presented by Kasai et al. (2012). In their work, Kasai et al. show that molecular oxygen, carbon monoxide, water (even heavy water), ozone, isotopologues of carbon dioxide, hydrogen peroxide, and various other hydrogen radicals all should have strong signals in the spectrum of Mars which they propose to observe. Additionally, wind speed parameters along the line of sight should have measurable readings. With this work we aim to develop the idea presented by Larsson et al. (2013) for remote measurement of magnetism by utilizing the Zeeman effect (Zeeman, 1897) on molecular oxygen in its ground state: $X(^3\Sigma_g^-)$. It is possible to combine this work with Kasai et al.'s idea into a single instrument that is capable of measuring and mapping both meteorological parameters and crustal magnetic structures, but this work will only focus on the magnetic aspects of flying such an instrument.

The Martian magnetic field is thought to be a remnant of a past global dipole that disappeared about 3.5 billion years ago (Acuña et al., 1998). All that remains of the past dipole is several magnetic sources in the crust, which were first measured in situ by the Magnetometer-Electron Reflectometer onboard the Mars Global Surveyor orbiter (Acuña et al., 1999) down to an





altitude of 100 km. The strongest sources are located in the southern hemisphere, with strengths of up to 2000 nT at 100 km altitude. Estimations of the strongest magnetic field strength at the surface based on these measurements vary from just above 10,000 nT, up to almost 20,000 nT (Brain et al., 2003). The ongoing Mars Atmosphere and Volatile Evolution mission will provide further coverage of the magnetic field down to ~150 km altitude in normal mode, but during week-long campaigns its

periapsis will be even lower at ~125 km altitude, improving the potential to map the crustal magnetism (Jakosky et al., 2015). There is to our knowledge one planned lander, InSight (http://insight.jpl.nasa.gov), which will carry a magnetometer to the equatorial surface of Mars.

The shape and distribution of the present field is informative for the crustal evolution of Mars (Nimmo and Tanaka, 2005). There are two main ideas about the magnetisms' formation: either a large impact demagnetized the north leaving the south

magnetized, or there was a southward migration of crustal material after the global dipole disappeared (see, e.g., Connerney et al., 2004; Citron and Zhong, 2012, for further discussions). Regardless of the reason, the strongest crustal field should be associated with the oldest intact crustal material since these regions are linked to the times when Mars still had an effective global dipole. Identifying the reasons why the northern hemisphere has a lower average elevation, and also a less magnetized crust, than the southern hemisphere is important for questions related to, e.g., how similar Mars and Earth were in their early

years. The method we propose for measuring the crustal magnetic field is useful in this regard, as it allows the determination of the magnetic field strength at different altitudes in the lower atmosphere where satellite magnetometers cannot reach. This means that it would help in the creation of profiles of magnetic field data that in turn can be used to estimate the depths, nature and locations of the crustal field sources. The estimation of these parameters is beyond the scope of the paper; the interested reader may check the work by Brain et al. (2003), Connerney et al. (2004) and references therein. However, we note from

Connerney et al. (2004) that there is more information about the structure of the magnetic field that is revealed at 100 km but is hidden at 400 km altitude (see their figure 6). If there are more structures at even lower altitudes, then these cannot be sampled by satellite magnetometers. Deep dips below 200 km altitude cost significant amounts of fuel due to increased air drag and thereby would shorten the lifetime of a satellite mission. Finding the crustal sources' characteristics accurately using only satellite data is thus difficult — which is why the available Martian magnetic field models differ in some regions by up

to 2000 nT (cf.  Cain et al., 2003; Morschauser et al., 2014) and the range of the strongest surface field is from just above 10,000 nT up to potentially 20,000 nT. The characteristics of the crustal field sources strongly limit the possible processes that led to the disappearance of the past dipole, to the north-south dichotomy, and to the strong crustal magnetic sources, making the crustal sources interesting targets for geological exploration.

## 2   Method

This work is based only on simulations. We perform limb simulations around the molecular oxygen absorption line at 368 GHz using a measurement scenario that achieves several measurements around the same latitude-longitude tangent profile during successive satellite revolutions. These simulated measurements are fed into a retrieval toolbox that estimates the errors of the magnetic field components in the tangent profile.



Section 2 is divided as follows. The first subsection summarizes the ideas behind the approach in generic terms. The second subsection goes over the basic aspects of the modeling theory. The last subsection describes the data required for the simulations and our practical design choices. Then Section 3 describes and discusses the results. Finally Section 4 serves our conclusion.

## 2.1   Measurement idea

The magnetic field influences molecular oxygen through the Zeeman effect. Molecular oxygen exhibits the clearest Zeeman effect in the sub-millimeter region of the molecules available in the Martian atmosphere. Other molecules have either a weaker Zeeman effect, or a lower volume mixing ratios, and are therefore less suitable for magnetic field retrievals. The Zeeman effect changes the energy states of the molecule to split an otherwise singular absorption line into several closely separated lines as a linear function of the strength of the magnetic field. The direction of the magnetic field is important as emission and absorption

are polarized by this splitting. The Martian crustal magnetic field is strong, but not strong enough to cleanly separate the split lines from the temperature and pressure broadening of the line shape. Measuring intensity peaks and valleys of the radiation on a frequency resolved grid to directly get the magnetic field strength by peak-to-peak frequency separation is not possible. Instead, the split lines act to broaden or shift the absorption profile by a few, up to some hundreds, of kHz. Such frequency broadening or shifting also happens from increased temperatures and from greater wind velocities.

So, an important question to ask is how can we distinguish the atmospheric effects from the magnetic effects on the absorption line? The most obvious way is to simply measure the polarization state of the radiation. Neither temperature nor wind polarizes the radiation so the level of polarization in the split/broadened spectra is from the magnetic field in clear-sky limb view. A full sampling of the polarization state of the radiation is thereby the best way to retrieve the magnetic field. However, it is more expensive and more difficult to build a sensor capable of measuring the different polarization components (so that

the components then can be combined for the total polarization state of the radiation), than it is to build a sensor capable of measuring just one polarization component. Assuming we can only measure one polarization component, is it still possible to get distinct magnetic information? Our approach has been to set up an observational strategy that observes the same limb tangent profile from several directions, multiple times during a few satellite revolutions. Temperature and pressure broadens the absorption line shape the same regardless of observational direction through the limb. Wind shifts the frequency along just

one axis. The total signal strength in a transparent passively emitting atmosphere is from the number density of the emitting molecule. What remains of the signal after these effects are accounted for is therefore the polarization state and frequency shift due to the Zeeman effect.

## 2.2   Theoretical considerations

We use the Atmospheric Radiative Transfer Simulator (ARTS; Buehler et al., 2005; Eriksson et al., 2011) to simulate mea-

surements. The retrieval toolbox **Q**-package (Qpack; Eriksson et al., 2005) is used to determine the magnetic field component error. Together, these code-bases set up our retrieval system.

The approach to radiative transfer taken by ARTS is to calculate the monochromatic pencil-beam polarized radiative transfer equation in Stokes formalism along the path of the radiation through a three-dimensional inhomogeneous atmosphere and



magnetic field. Antenna size, sensor characteristics, and polychromatic signal averaging are considered (as described by Eriksson et al., 2006). The Zeeman effect module of ARTS (Larsson et al., 2014) is applied for these calculations to simulate the left circular polarization component as observed by the simulated sensor for a 20 MHz passband of 201 channels of 100 kHz Gaussian shape surrounding the central absorption line at 368 GHz. Circular polarization is noticeably more influenced by the

magnetic field than linear polarization, and 10 MHz on both sides of the line capture most of the information given by the Zeeman effect. For circular polarization, the magnetic signal is strongest when the local magnetic field vector points directly at or is pointing directly away from the sensor (along the track of the measured radiation). So in limb viewing geometry radiative transfer, only the horizontal magnetic field components are important at the tangent point. We expect some sensitivity to the radial component at higher altitudes than the tangent altitude (since part of the radial component is then along the line

of sight), but the sensitivity to the radial component is expected to be low since most of the signal is from around the tangent point. If we measure linear instead of circular polarization, then these measurements are sensitive to both the radial component and the horizontal components of the magnetic field. However, the magnetic signal significantly weaker for linear than for circular polarization so the retrieved magnetic field would be noisier. The ARTS simulations also give the Jacobian matrix for specified retrieval quantities (in our case the magnetic field components and, for testing, temperature, wind components, and

the molecular oxygen volume mixing ratio).

We use a moderately non-linear error characterization method, as presented by Rodgers (2000), to estimate the errors associated with a simulated measurement on a retrieval quantity. This error characterization is from

$$\mathbf{x} = \mathbf{x}_a + \mathbf{S}_a \mathbf{K}^\top \left( \mathbf{K} \mathbf{S}_a \mathbf{K}^\top + \mathbf{S}_\epsilon \right)^{-1} \left[ \mathbf{y} - \mathbf{F}(\mathbf{x}) + \mathbf{K} \left( \mathbf{x} - \mathbf{x}_a \right) \right], \tag{1}$$

where $\mathbf{x}$ is the derived atmospheric variables and magnetic field, $\mathbf{x}_a$ is the a priori atmospheric variables and magnetic input,

at different retrieval grid points (atmospheric pressure level, latitude, longitude), $\mathbf{S}_a$ is the covariance matrix of the a priori, $\mathbf{K}$ is the Jacobian matrix, $\mathbf{S}_\epsilon$ is the covariance matrix of the instrument error, $\mathbf{y}$ is the simulated measurement vector (made up of several individual measurements), and $\mathbf{F}(\mathbf{x})$ is the forward model simulations. With measurements $\mathbf{F}(\mathbf{x}) \neq \mathbf{y}$, however with pure simulations the terms are identical. Therefore, the error itself is found from $\mathbf{F}(\mathbf{x}) = \mathbf{y} - \epsilon$, where $\epsilon$ is the random noise of the observation — this 'noise' encompasses every error not accounted for by the simulated measurement especially defined

instrumental errors. Thus the retrieval error of Equation 1 is $\mathbf{S}_a \mathbf{K}^\top \left( \mathbf{K} \mathbf{S}_a \mathbf{K}^\top + \mathbf{S}_\epsilon \right)^{-1} \epsilon$. We also estimate the smoothing of the calculations by the matrix $\mathbf{S}_a \mathbf{K}^\top \left( \mathbf{K} \mathbf{S}_a \mathbf{K}^\top + \mathbf{S}_\epsilon \right)^{-1} \mathbf{K}$, which is called the averaging kernel. This matrix gives information on the measurement response of the system and the vertical resolution of the measurements. The method above gives a linear error estimate; even if a problem is inherently non-linear, the error is often still linear (Rodgers, 2000).

About the retrieval grid, as explained in the previous subsection, several measurements of radiation from the same tangent

profile can be combined to find the magnetic field components. The observational geometry is important as the individual measurements will observe different parts of the atmosphere at higher altitudes but the same parts of the atmosphere at lower altitudes. We therefore use a retrieval grid that takes the three-dimensional inhomogeneous atmosphere and magnetic field into account by setting a 3-by-3 grid of latitudes and longitudes, with the central grid point at the latitude and longitude of the tangent profile. The latitude grid positions are separated by a change of 2 degrees, with the same horizontal distance separating




the longitude grid positions. We use a vertical retrieval grid separation of 2 km from 0 km up to 100 km. So $\mathbf{x}_a$ is a $3 \times 3 \times 51$ long vector of inputs per retrieval quantity. We note that the measured magnetic field for the altitude range we have considered (below 100 km) is essentially the planetary crustal magnetism (Brain et al., 2003). External fields (interplanetary magnetic fields draping around the conductive ionosphere and induced ionospheric magnetic fields) play a role at higher altitude where

the solar wind interacts with the upper atmosphere. In areas of significant crustal magnetism, the influence of external fields starts at altitudes around 500 km (figure 11 of Brain et al., 2003), therefore they are not a concern here.

For $\mathbf{S}_a$, we assume that there is no correlation between the different retrieval quantities but that there is some correlation in spatial distance. For the tests we have performed on the wind, temperature and volume mixing ratio retrievals, we use covariance matrices that works for Earth. For the magnetic field components, since the crustal magnetic field is a mostly static

variable at the altitudes below 100 km where we simulate the measurements, its covariance matrix should also be mostly static. However, in a Bayesian framework, the covariance matrix should describe the knowledge we have of the magnetic field at the time of measurement. As mentioned earlier, the strongest differences between models in Morschauser et al. (2014) was 2000 nT but many sources differ less. So different models give different results. For simplicity we set the uncertainty of the magnetic field at 1000 nT at all altitudes, and that the correlation in altitude is $e^{-1}$ after one order of magnitude pressure

change. We also set a small correlation between horizontal grids of $e^{-1}$ after a horizontal distance equivalent to 1.5 degrees of latitude change.

## 2.3   Model inputs

### 2.3.1   Orbit and sensor considerations

A circular orbit at 330 km altitude with around 2 hour period and 97° inclination and ascending node at 0° is used in all

simulations. This orbit was selected ad hoc, such that it is able to cover Mars in a short time frame with measurements covering almost a full circle in azimuthal resolution. Other orbital parameters can be used to retrieve the magnetic field as well but require different considerations. An elliptical orbit, for instance, will have different vertical resolutions depending on true anomaly, and a less inclined orbit means that a different measurement scheme has to be adopted and that less polar-ward latitudes can be sounded — most likely with increased measurement noise. With enough measurements the measurement noise could be

reduced regardless of orbit. However, the details of the suggested measurements to any specific orbit has to be worked out on a per-orbit basis, so how much the noise can be reduced is not possible to tell without doing specific simulations for a given set of orbital parameters.

With this choice of orbital parameters, we model a sensor with a 30 cm diameter antenna for a vertical resolution of about 4 km at the tangent points sampled in limb sounding by an orbiter at 330 km altitude. The antenna size was chosen for these

simulations, because it is reasonably small. We will only simulate left circular polarized radiation and we assume 1500 K single sideband system noise temperature. This number is a rough extrapolation to lower frequencies from the expected system noise temperature of the Jupiter Icy Moons' Sub-millimeter Wave Instrument's 600 GHz band (see Sobis et al., 2011, 2014; Treuttel





et al., 2016, for more details on the JUICE/SWI receiver). We enforce 2 seconds as the time of a measurement, dedicating one third to calibration efforts and the remaining ∼1.3 seconds as integration time.

The position of the tangent profiles tracks this orbit as a pseudo-orbit with the same orbital parameters but with an ascending node offset by half the longitude drift of a full satellite revolution. These tangent profiles can be observed in limb geometry
four times within about two revolutions; for each revolution: once before passing the tangent profile and once after passing the tangent profile. The tangent altitudes we select to observe this tangent profile extend from 6 km to 78 km altitude, with a 3 km vertical separation for 25 tangent altitudes. So a total of 100 individual measurements are considered (25 measurements × 4 observation sets) in the error characterization. We call the grouping of all these individual measurements for a given tangent profile a measurement block. The error characterization of this work is from these measurement blocks. The results in the
next sections track the tangent profile pseudo-orbit through 7 complete revolutions at a distance of 15 degrees of true anomaly between two retrieval profiles. This is a sufficient number of revolutions so that we can give the global error characterization.

An example of observation geometry and orbit can be seen in Figure 1 (the supplementary material provides additional details). As mentioned, we simulate 201 channels for the sensor. With 100 individual measurements in a measurement block, $\mathbf{y}$ is a 20,100 long vector and $\mathbf{K}$ is a $20{,}100 \times (3 \times 3 \times 51 \times n)$ large matrix, where $n$ is the number of retrieval quantities (usually
three for the magnetic field components) and the other numbers are from the retrieval grid. The position of all 100 of these individual measurements in the example of Figure 1 is shown in panel (a). Panel (b) of the same figure shows the grouping of individual measurements for one set of observations of the tangent profile. We call this smaller grouping a measurement cluster. A satellite moving towards a tangent profile will, by geometrical necessity, see the tangent altitude increase the closer it gets to the profile, and vice verse a satellite moving away will see the tangent altitude decrease the farther away it gets. Finally,
panel (c) shows that there is a slight drift in tangent point positions between different measurements. We enforce a 2 seconds measurement time, and we enforce constant tangent altitudes. It is thus expected that the tangent position will drift slightly; the greater circle of positions that can observe a tangent altitude does not strictly intersect the satellite orbit at discrete time intervals. The drift is accounted for in the calculations of $\mathbf{K}$ and $\mathbf{y}$.

The only spectroscopic effect included in our radiative transfer calculations is the contribution of the 368 GHz molecular
oxygen absorption line. We select this absorption line, because the Zeeman effect line splitting frequency is linear in magnetic field strength. So the lower the frequency, the stronger the signal from the magnetic field becomes as Doppler broadening is also linear with absorption line frequency. The reason we are not simulating one of the 60 GHz molecular oxygen band lines is because we consider the antenna diameter required to have a decent vertical resolution in limb view as too large to reasonably fly the sensor. Our line data is from a recently compiled planetary toolbox, which gives line pressure broadening as a function
of atmospheric composition for mixtures of common planetary species[1] by Mendrok and Eriksson (2014) who assume that the molecular oxygen pressure broadening by carbon dioxide is ∼20 kHz/Pa at 296 K for our absorption line. This is directly taken from the 118 GHz line value presented by Golubiatnikov et al. (2003). This spectroscopic toolbox allows for direct calculations in Mars' carbon dioxide rich atmosphere. Note that collision-induced absorption between pairs of carbon dioxide molecules is also important for the total absorption in our frequency range. Using the model by Ho et al. (1971), we estimate that it can be

---

[1]Our atmospheric and spectroscopic data are available in Extensible Markup Language files designed for ARTS via www.radiativetransfer.org.





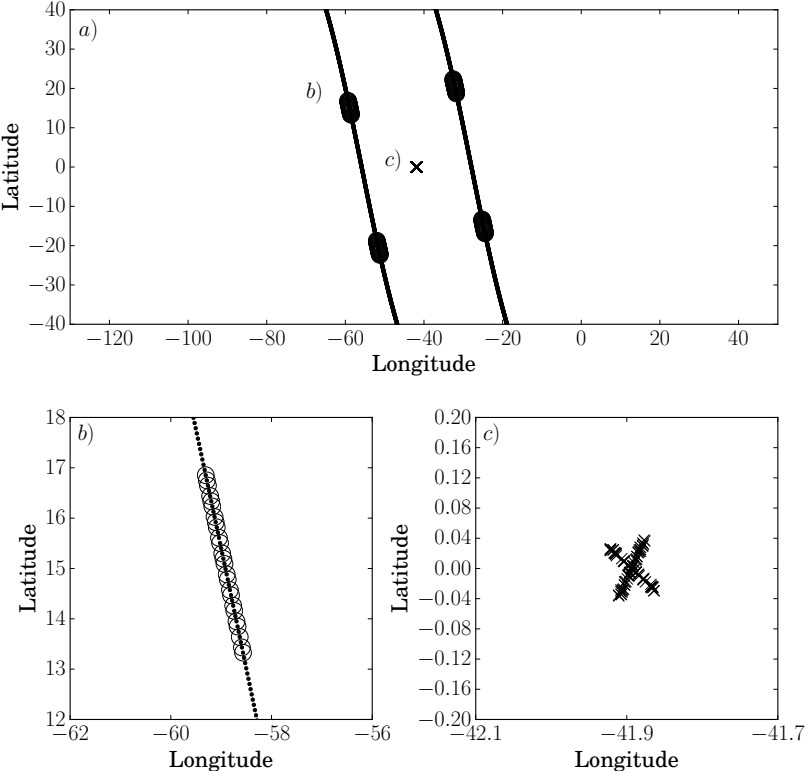

**Figure 1.** Example of the geometry of a measurement block. Panel (a) shows the orbit of a satellite as 2 thin lines indicating 2 satellite revolutions; the satellite positions during the 2 observation sets during each revolution in the measurement block as larger circles, and the mean tangent position of the measurement block as a cross. This example retrieval point is above a location of moderate modeled magnetism strength near the equator (750 nT at surface altitudes crustal magnetism model by Cain et al., 2003). Panel (b) zooms in to show the satellite positions at one of the clusters of satellite positions. Panel (c) zooms in to show the tangent positions. The original intent of the measurement block was to observe in the center of this block of individual measurements.

as important as adding a 20 K baseline signal for a pencil beam tangential to the surface. This added absorption is not sufficient to make the atmosphere opaque. We can anyways ignore this effect because at 6 km altitude — our lowest tangent altitude — the collision-induced absorption is much weaker than at the surface. Still, we note this as a lacking feature to be added in the future.

5    To summarize orbit and sensor constraints, we present Figure 2 to give an overview of how a simulation of a measurement block looks like with the assumptions outlined above. Panel (a) contains peak brightness temperature measurements for 2 clusters (looking ahead and behind the tangent point) for one revolution, and again measurements for 2 clusters for another revolution, with each cluster having 25 measurements at tangent altitudes between 6 km and 78 km. As expected, the main contribution to the signal strength is the tangent altitude (through which the column number densities are regulated). At 6 km

10   measurement tangent altitude, the radiation signal has a peak brightness temperature of about 140 K, and at 78 km the same




number is about 10 K. The sinusoidal pattern of measurements in each cluster comes from the resolution of the spectra: the peaks are at the line centers and the valleys are away from the line centers. Only a very small part of this signal is influenced by the magnetic field. This part of the signal is not clear from the brightness temperature in panels (a-b) but is instead shown in the Jacobian panel (c). We see that the magnetic signal is of the order of about 0.2 mK/nT as strongest per 2 km altitude level

per 100 kHz spectrometer channel bin. The resolution of the antenna is indicated from the coverage of the circular central disk of the Jacobian panel. Again, the simulated full-width vertical resolution is around 4 km for a 30 cm antenna. The covariance matrix is shown for completeness in panel (d). The figure neither shows the full $\mathbf{K}$ nor the full $\mathbf{S}_a$ matrices but is zoomed in on a single sub-matrix. Nevertheless, the figure presents the basic retrieval setup. The only missing entry required for the error characterization is $\mathbf{S}_\epsilon$. This matrix is not shown because it is simply filled with a constant describing the noise of the

sensor. This constant is the square root of the system noise equivalent temperature divided by the integration time times the spectrometer channel full-width — which turns out to be just below 0.1 K$^2$.

### 2.3.2 Atmosphere and magnetic field models

The molecular oxygen profile is assumed constant at a volume mixing ratio of 1400 parts per million volume, which follows the profile reported by Hartogh et al. (2010b). This profile was derived by observations of the HIFI instrument on Herschel as

part of the Herschel Solar System Observations program (Hartogh et al., 2009) by using a temperature profile derived at the same day from carbon monoxide observations (Hartogh et al., 2010a). At altitudes above 90 km, this constant mixing ratio is not valid (as shown by measurements presented by Sandel et al., 2015), and it is likely not valid at our lower altitudes either. However, since we can see the molecular oxygen radiometric signal even at 78 km altitude, and since a changed mixing ratio only affects the total signal strength, the magnetic signal — that is the relative polarization and splitting caused by the magnetic

field — is not affected by this different mixing other than lowering or increasing the sensitive altitude range. With this selection of volume mixing ratio, we are not sensitive to molecular oxygen at altitudes much higher than 78 km. This is seen by the low signal strength at this altitude in Figure 2 panel a, and the reduction compared to lower tangent altitudes. Additionally, panel c of the same figure shows that most of the signal is from around the tangent point, which is a property of limb observation geometry. Higher altitude molecular oxygen content is therefore not important for our results.

We use the northern spring (solar longitude $L_s = 0°$) carbon dioxide volume mixing ratio, temperature profile, wind profile, and pressure profiles from the ARTS planetary toolbox by Mendrok and Eriksson (2014), who base it on the Laboratoire de Météorologie Dynamique's Global Circulation Model [LMD GCM] by Forget et al. (1999). These atmospheric profiles only provide global averages during the season. We thus ignore most effects of time since we are interested in demonstrating the feasibility of the technique of magnetic field measurements on a general basis. Presented errors are hence of average character.

The a priori atmospheric profiles can be constrained by the measurements but we have not taken this into account in any of our simulations shown in this work, though we give an example of what adding the atmospheric components to the error characterization means for the errors of the magnetic components. If the atmospheric parameters are not stable over the 2 hours of a revolution, then the evolution of the temperature and wind profiles must be accounted for in the retrieval setup. This is still possible without changing the theoretical formalism of the retrieval setup, if the problem remains moderately non-linear,







**Figure 2.** Example retrieval scenario. This is **y**, part of **K**, and part of $\mathbf{S}_a$ from the example in Figure 1. Panel (a) is the simulated measurements **y** of ARTS with individual clusters marked by their revolution number and looking direction. Each peak corresponds to an individual measurement — the horizontal axis gives the tangent altitudes. The bottom row zooms in on the marked region of **y** to show an individual simulation at 48 km tangent altitude (panel b), a zoom on the transpose of the **K** sub-matrix at the tangent point latitude-longitude grid point for one magnetic component (panel c), and its corresponding $\mathbf{S}_a$ sub-matrix (panel d).

but it complicates the preparation of the atmospheric data and the simulations. From a theoretical point of view, the error characteristics of the magnetic field will remain moderately non-linear even if the temperature, wind, and volume mixing ratio are accounted for in the retrieval problem because there should be no correlation between the atmospheric parameters and the magnetic field.

5     The primary magnetic field in our altitude range is the crustal magnetic field (Brain et al., 2003). Our model magnetic field is from the spherical harmonics fit to a selection of Mars Global Surveyor crustal magnetic field vector data by Cain et al. (2003). The magnetic field in ARTS is an extraction of Cain et al.'s fit that has been gridded at a global resolution of 0.5 degrees latitude, 0.5 degrees longitude, and 5 km altitude from the surface up to 100 km altitude. This means that the magnetic




field is allowed to change along the path of the radiative transfer in our simulations. If the magnetic field had changed more dramatically through the radiative transfer than it does, then this might have reduced the magnetic signal by changing the polarization of the signal propagation. Since this is not a problem that we encounter, we will not pursue any ideas on how to deal with it. The model by Cain et al. (2003) is based on a ninetieth order Legendre polynomial, so the spatial resolution

is limited to ∼30 km. Other models, such as Morschauser et al. (2014), differ from Cain et al.'s by about 2000 nT at most (see figure 9 of Morschauser et al., 2014), but are otherwise close. This 2000 nT difference is therefore an estimation of the accuracy of the magnetic field components with the present data. Some authors (Brain et al., 2003) speculate that the strongest field strength at the surface is upwards to 20,000 nT. Cain et al. give the strongest field as 12,000 nT, so 8000 nT serves as an estimate of the potentially largest discrepancy today of the crustal magnetic field strength of Mars.

## 10   3   Results and discussions

The structure of this section is that we begin by presenting the results of running the measurement block of Figure 1. This example is used as a generic measurement block and noise levels are lower or higher depending on the geometry of observations. We then present an average for many observations spread out globally. There is a final summary at the end of this section showcasing where our results suggest that the suggested measurements would be useful given the expected noise levels

of individual measurement blocks. Note that we define the magnetic field components as: $B_u$ is the east-west component of the magnetic field, $B_v$ is the north-south component, and $B_w$ is the radial component.

### 3.1   Example measurement

Figure 3 shows the results of running the retrieval system for the tangent profile of the measurement block in Figure 1 with the simulated measurements and Jacobian as in Figure 2. It shows that the measurement response is good (i.e., around unity)

for the $B_u$ and $B_v$ components (horizontal components) from the ground up to about 70 km. The $B_w$ component (vertical component) has a poorer response than the other two components, but the measurements are in this case sensitive to $B_w$ at altitudes around 40 km. The vertical resolution is about 4 km for $B_u$ and $B_v$ over the entire sensitive altitude range because it is perpendicular to the line of sight in the limb geometry, which reflects the achievable resolution for the simulated antenna size and tangent profile altitude spacing. The vertical resolution is much worse for $B_w$ between 8 km and 15 km. The observational

error that $B_u$ and $B_v$ experience varies between around 150 nT to 200 nT in the sensitive altitude range. The observational error that $B_w$ experiences is much worse, but it settles close to 300 nT at sensitive altitudes. Note that the observation errors are low where the measurement response is low; without measurements there are no errors due to observations.

As for vertical structures, the main limitation at lower altitudes is that pressure broadening hides the magnetic signal, and the main limitations at higher altitudes is that there is very little molecular oxygen due to the lower pressure. The optimal magnetic

signal is from around 40 km, which is why the noise at these altitudes is lower (panel b).

$B_w$ is more difficult to detect than the other two components because of the characteristics of the Zeeman effect. The multiple measurements from different azimuthal directions allow the angle of the magnetic field in the horizontal plane to be





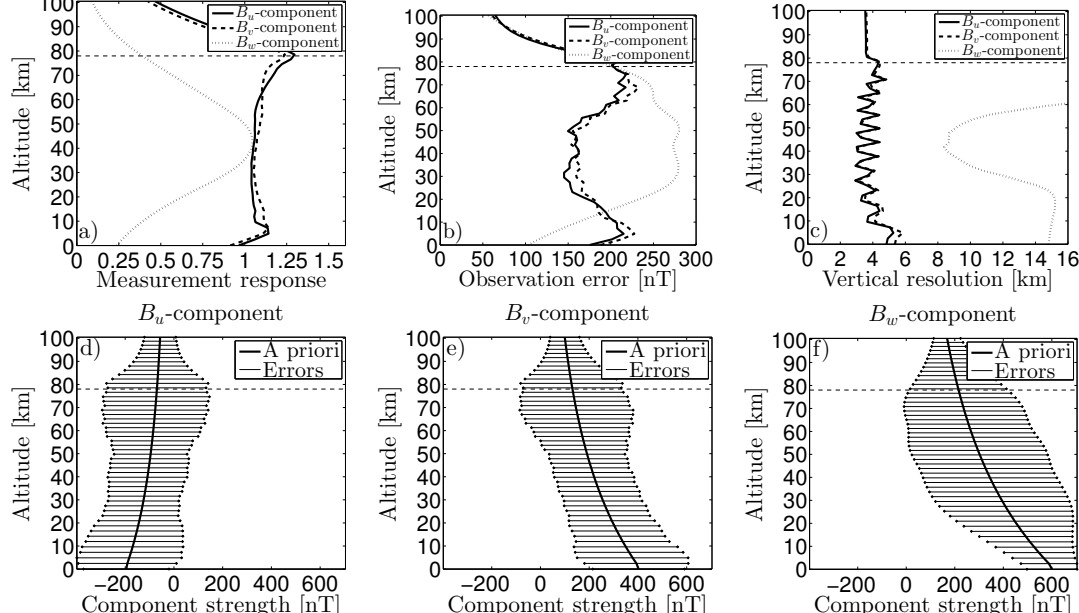

**Figure 3.** Retrieval information for Figure 1. The first row shows Qpack output following Equation 1. Panel (a) shows the response function per magnetic field component, panel (b) shows the observation error per magnetic field component, and panel (c) shows the vertical resolution per magnetic field component (zig-zag is from discrete number of vertical limb measurements at 3 km spacing). The lower row (panels d-f) compares this observational error to a priori magnetic field components. The horizontal dashed line shows the highest tangent altitude.

tested almost directly, but $B_w$ acts only to broaden the absorption line in limb view. It is therefore difficult to measure it without more constraints. In fact, if we add retrievals of wind, temperature and molecular oxygen volume mixing ratio to the retrieval problem — thus reducing the constraints on the retrieval problem — the measurement response to $B_w$ is no longer good at any altitude. So we will ignore $B_w$ from here on since it is not measurable by the presented method. The horizontal magnetic

5   components are not affected by the lessened constraints on the atmospheric parameters.

## 3.2   Suggested operational setup

The average observation errors expected for our suggested operational setup scenario are presented in Figure 4. We simulate only 7 revolutions, or about half a Martian day, of measurements, since this gives a good coverage for the whole Martian globe with our ad hoc orbit.

10   We find that the global average profile observation error for the $B_u$ component is 170 nT and that the same average observation error for the $B_v$ component is 200 nT. Near the equator, the two components have similar observation errors (almost identical to what is shown in Figure 3). Closer to the poles, $B_u$ has better observation errors, and $B_v$ has worse observation errors. The reason for this is the azimuthal angles of the observation geometry of a measurement block. With the selected orbit and observational strategy, the tangent profiles are observed from the north-east/west and south-east/west azimuthal angles at





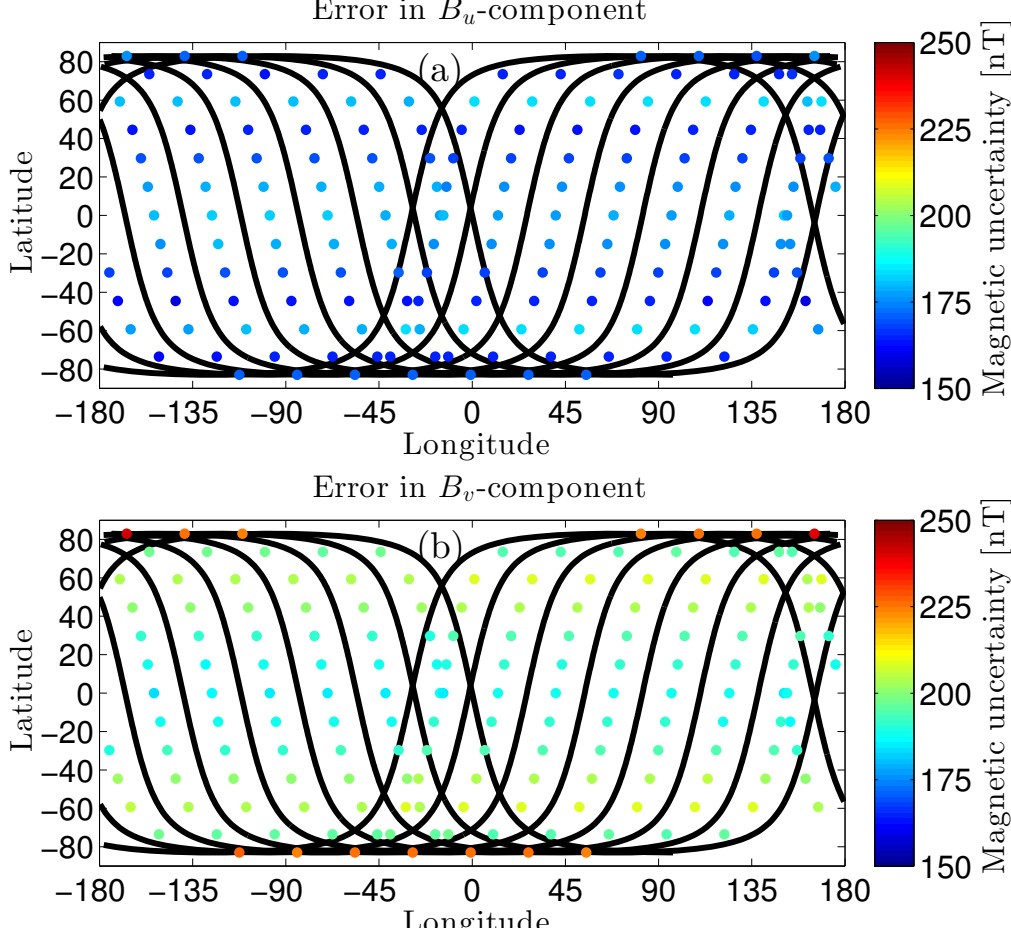

**Figure 4.** Suggested operational setup scenario. The black lines represent the orbit through seven revolutions. The maps show the estimations for errors per component averaged between 6 km and 78 km altitude.

the equator. Near the poles however, the azimuthal angles are almost aligned with the east-west direction. This indicates that it is possible to improve $B_v$ observation errors at polar latitudes to the same levels as $B_u$ by more observations from directly north or south of the tangent profiles. Similarly, it also indicates that it is possible to improve the equatorial observation errors for both horizontal components to the levels of $B_u$ at polar latitudes by focusing on east-west or north-south observation
5  geometry depending on the component of interest.

### 3.3 Simple estimation of measurable area and a comparison to satellite magnetometer

We prepared Figure 5 to show the areas over the Martian surface where these measurements would be useful. This plot shows the modeled magnetic field strength at 40 km altitude from Cain et al. (2003). It turns out that an area covering about 36% of Mars total area has magnetic field strengths above 200 nT, since the magnetism is not homogeneous but localized, with


the southern sources being stronger than the northern sources. The areas with more than 1000 nT field strength cover about 4% of the surface. So our suggested measurements are estimated to provide information on the Martian magnetic field over somewhere between 4% and 36% of the area of Mars; i.e. in areas where the crustal magnetism is significant.

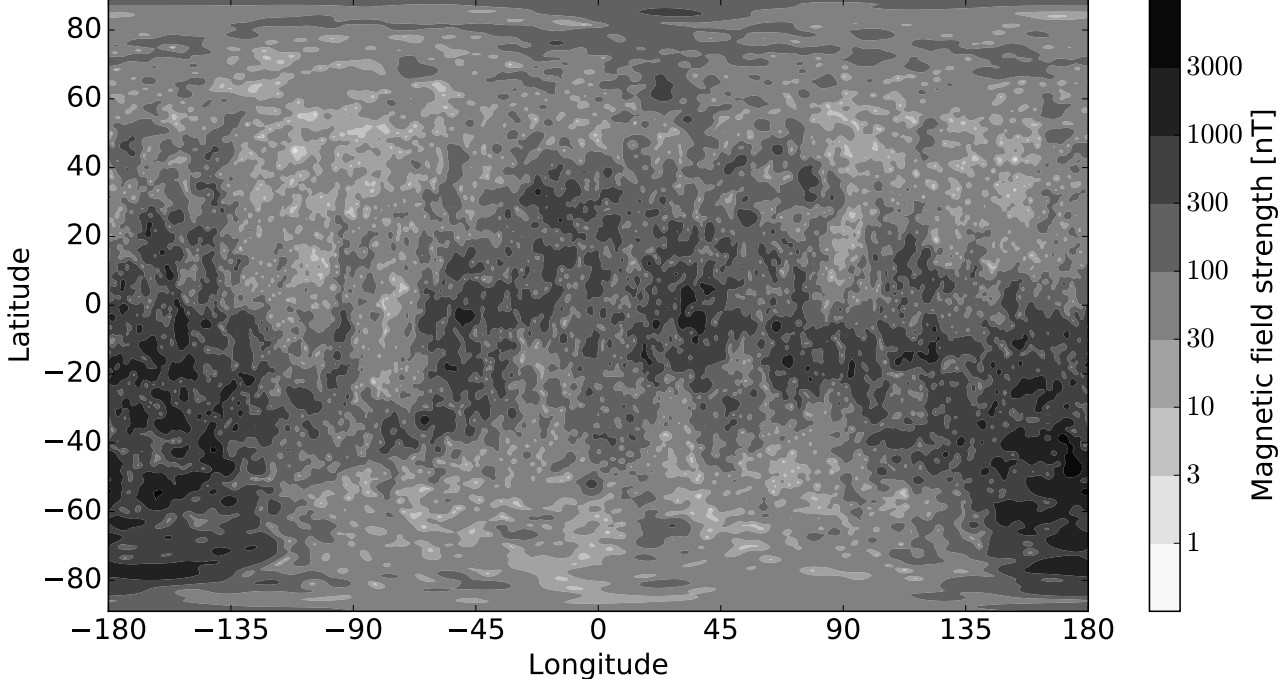

**Figure 5.** The modeled magnetic field strength of Mars at 40 km altitude from Cain et al. (2003).

Comparing the 200 nT noise attainable by sub-millimeter measurements to satellite-borne magnetometer measurements by
5   Mars Global Surveyor offers some additional information into the limitations and possibilities of the sub-millimeter measurements. The MGS magnetometer had noise levels of only 3 nT during its deep dips to ∼100 km (Acuña et al., 1998). At 100 km, almost 99% of the area of Mars has a magnetic field strength stronger than 3 nT, with 76% of the area having a magnetic field strength over 15 nT (Cain et al., 2003). For the same relative noise levels the sub-millimeter measurements covers only 4-36% of the area. How much area that a satellite-borne magnetometer can cover at 100 km altitude during a mission is therefore
10   mostly limited by the amount of fuel brought with the satellite to maintain a good orbit after the deep dips. Measurements far above 100 km altitude contribute less to understanding the structures of the magnetic field necessary to estimate the strengths, shapes and locations of the crustal sources. This is because the structures of the magnetic field are hidden by the added distance (as seen in, e.g., figure 6 by Connerney et al., 2004). The higher altitude of these more stable orbits are thus less interesting in regards to crustal sources but nevertheless offer much better noise levels. The noisier sub-millimeter measurements are able to
15   sense lower altitudes and, in contrast to the magnetometer, this can be done from a stable orbit. We consider that the measurements of the satellite-borne magnetometers and the sub-millimeter sensors are complementary to one-another, with satellite





magnetometers able to more strictly limit the strength of the field and sub-millimeter spectrometers more capable to sample low altitude structures.

## 4   Conclusions

We describe an idea that makes measurements of the Martian near-surface crustal magnetic field profiles possible using remote sensing techniques by combining several limb observations. We have performed radiative transfer simulations for radiation measurements in the atmosphere of Mars for one orbiting sensor. These radiative transfer simulations have been fed into a retrieval toolbox to find sensitivities to the magnetic field. Our work shows that limb observations of the sub-millimeter radiation of one absorption line of molecular oxygen can be used to measure two of the three components (the horizontal components) of the magnetic field with around 200 nT accuracy. The vertical resolution of such measurements will be about 4 km for a 30 cm diameter antenna measuring the 368 GHz line from 330 km altitude. The described measurements are sensitive to the magnetic field strength over about one-third of the Martian surface, which is the area where significant localized planetary magnetic fields exist. We have made few assumptions on correlations in the retrieval system to let our method enhance measurement sensitivities rather than a priori sensitivities. With reasonable assumptions on the magnetic source fields or with different orbit/observation geometry, it should be possible to reduce the noise further. We suggest flying a sensor capable of measuring the molecular oxygen absorption line at 368 GHz on a satellite mission to Mars to make a detailed map of the Martian crustal magnetic field and thereby help — in combination with available satellite magnetometer data — to find the depths, shapes and positions of the crustal field sources.



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
