# Peer review of "Martian magnetism with orbiting sub-millimeter sensor: Simulated retrieval system"

_Geoscientific Instrumentation, Methods and Data Systems, 2016_

## Referee Comment (RC1) · Anonymous Referee #2 · 28 Sep 2016

Review of: Martian magnetism with orbiting sub-millimeter sensor: Simulated retrieval system, by Larsson et al. submitted to GI.

Summary: The paper gives a description of a simulation of the crustal magnetic field of Mars (6-70 km altitude) based on the Zeeman effect of the molecular oxygen absorption line (368 GHz). The work is totally based on simulations with no comparison (quantitatively speaking) to real data from Mars or previous models. Finally, authors state that with this technique, the crustal magnetic field between the 4 and 36 % of the Mars surface could be roughly estimated. I appreciate the efforts made by the authors to improve the paper to be published in the discussion section. The paper is interesting and after second revision, I consider the paper needs minor revisions to be addressed with this open discussion.

[Figure]

Discussion:

1. Figure 6 is not very clear in a grey color scale. Would you mind to change it into a color scale and highlight the 4-36% of the surface to be sampled by this technique with a dashed line? This magnetic field map changes with altitude, so would you mind to show in the paper 3-4 panels with this map at different altitudes and with the percentage of surface sampled at each altitude?

2. The error in the horizontal magnetic components (∼200 nT) is of the same order of the component strength (Figure 3), and this is after considering ideal conditions for this simulation. I would like to see a larger discussion about this fact and how a different orbit configuration would affect that.

3. What's the error of the Cain model at the low altitudes of this paper?

4. ARTS simulator should be briefly described in the text, despite the references.

Minors:

5. Lines 6-7 of second page: ExoMars 2020 surface platform also plans to carry a magnetometer to the surface of Mars, called: MAIGRET.

6. Line 14 of second page: you should add that the Northern Hemisphere is much younger than the Southern Hemisphere.

7. Line 25 of page 5: do authors mean "measurements of the measured noise"?

8. Answer given to referee about the JUICE sideband should be added to the paper.

Good luck!

---

## Referee Comment (RC2) · Anonymous Referee #3 · 5 Oct 2016

**Geosci. Instrum. Method. Data Syst.**

Manuscript Number: doi:10.5194/gi-2016-12, 2016

Title: **Martian magnetism with orbiting sub-millimeter sensor: Simulated retrieval system**

Authors: Richard Larsson, Mathias Milz, Patrick Eriksson, Jana Mendrok, Yasuko Kasai, Stefan Alexander Buehler, Catherine Diéval, David Brain, and Paul Hartogh

The manuscript is well-written and I consider that the work can be considered for its publication.

In general terms I consider that there is still much work to be able to produce results with enough sensitivity using this method (at least with the sensitivity of the already done measurements: Mars Global Surveyor). Maybe one of the major inconveniences of the manuscript is the example of application. Also the work is based on the oxygen concentration determined by the models and this can have some errors, as well as inhomogeneities and temporal variations.

In particular I suggest the following modifications based on the content:

It is said in 2.1, line 18, that "A full sampling of the polarization state of the radiation is thereby the best way to retrieve the magnetic field."

On the one hand the statement should be moderated since the works done with previous missions carrying magnetometers have casted better results than this theoretically method. Also the success in its application on Mars precisely is very doubtful. (The error is in the order of magnitude of the range). On the other hand, only one component of the polarized radiation is retrieved. Regarding the sources that may contribute to this single component, it is assumed some approximations that constrain these contributions. However, without these approximations, would the final value of 200 nT error in the magnetic field sensitivity be affected?

If it is the case what would be the impact on the global coverage ( < 4 % or < 36 % ?).

It would be very important to make a connection with the utilization of this methodology in an application with better results. (It can be with atmospheric examples but the discussion would have to be included in the manuscript).

And the following regarding the style:

1) The manuscript should be reviewed to avoid repetitions. For instance: page 2 line 2 and in the same page line 25, and in page 3, line 16. All the paragraph is repeated. Additionally, the question mark might be removed.
2) Abstract. Lines 1 to 5, Rephrase. Suggestion: "A Mars-orbiting sub-millimeter sensor can be used to retrieve the magnetic field at low altitudes over large areas of significant planetary crustal magnetism of the surface of Mars from measurements of circularly polarized radiation emitted by the 368 GHz ground-state molecular oxygen absorption line".

3) In 1, page 2, line 25, is repeated what is said in the line 3, and it is repeated in the page 10 line 8. This should be rephrased to avoid repetition.
4) Page 2, line 15, shouldn't have the interrogation at the end of the sentence. Lines 15 to 18 should be rephrased to avoid repetition.

END OF REPORT

---

## Author Comment (AC1) · 5 Dec 2016

The attached zip-file contains an updated version of the manuscript. All relevant files are attached. There is no update in the supplementary material of supplementary.pdf. The main.pdf file contains the main manuscript and diff.pdf is an automatically generated difference file between this version and the previously submitted version of the manuscript where blue color text marks new text and red color text marks removed text. Finally, point-by-point response to the comments by the reviewers are given in answers.pdf, where the comments are marked by indented italic text and the responses are in plain text. Please see main manuscript for references found in answers.pdf

Please also note the supplement to this comment:

[Figure]

http://www.geosci-instrum-method-data-syst-discuss.net/gi-2016-12/gi-2016-12-AC1-supplement.zip